# Angiotensin II-Induced Signal Transduction Mechanisms for Cardiac Hypertrophy

**DOI:** 10.3390/cells11213336

**Published:** 2022-10-22

**Authors:** Sukhwinder K. Bhullar, Naranjan S. Dhalla

**Affiliations:** Institute of Cardiovascular Sciences & Department of Physiology & Pathophysiology, Max Rady College of Medicine, University of Manitoba, Winnipeg, MB R2H 2A6, Canada

**Keywords:** Ang II-induced cardiac hypertrophy, Ang II-induced signal transduction, AT_1_ receptors, AT_2_ receptors, mass receptors, Ca^2+^-overload and calcineurin, oxidative stress

## Abstract

Although acute exposure of the heart to angiotensin (Ang II) produces physiological cardiac hypertrophy and chronic exposure results in pathological hypertrophy, the signal transduction mechanisms for these effects are of complex nature. It is now evident that the hypertrophic response is mediated by the activation of Ang type 1 receptors (AT_1_R), whereas the activation of Ang type 2 receptors (AT_2_R) by Ang II and Mas receptors by Ang-(1-7) exerts antihypertrophic effects. Furthermore, AT_1_R-induced activation of phospholipase C for stimulating protein kinase C, influx of Ca^2+^ through sarcolemmal Ca^2+^- channels, release of Ca^2+^ from the sarcoplasmic reticulum, and activation of sarcolemmal NADPH oxidase 2 for altering cardiomyocytes redox status may be involved in physiological hypertrophy. On the other hand, reduction in the expression of AT_2_R and Mas receptors, the release of growth factors from fibroblasts for the occurrence of fibrosis, and the development of oxidative stress due to activation of mitochondria NADPH oxidase 4 as well as the depression of nuclear factor erythroid-2 activity for the occurrence of Ca^2+^-overload and activation of calcineurin may be involved in inducing pathological cardiac hypertrophy. These observations support the view that inhibition of AT_1_R or activation of AT_2_R and Mas receptors as well as depression of oxidative stress may prevent or reverse the Ang II-induced cardiac hypertrophy.

## 1. Introduction

It is now well-known that the activation of both circulating and local cardiac of renin-angiotensin system (RAS) results in the release of angiotensin II (Ang II), which exerts powerful effects on the cardiovascular system [1,2,3,4,5,6,7,8,9,10]. The activation of RAS normally occurs as a consequence of a fall in blood pressure and/or an increase in ventricular wall stress as a compensatory mechanism to maintain hemodynamic homeostasis. At early stages, Ang II has been demonstrated to increase blood pressure and produce positive inotropic effect in addition to inducing growth of the myocardium (adaptive cardiac hypertrophy) and promoting angiogenesis [11,12,13,14,15,16,17,18,19,20,21,22,23,24]. However, the exposure of the heart to Ang II for a prolonged period results in the transition of adaptive or physiological cardiac hypertrophy into maladaptive or pathological cardiac hypertrophy, which is considered to serve as a risk factor for the development of heart failure [25,26,27,28,29,30]. Several other vasoactive hormones and different interventions such as pressure overload and volume overload have also been reported to induce both physiological and pathological cardiac hypertrophy [31,32,33,34,35,36,37,38,39,40]. These observations from various experimental models including those due to elevated levels of circulating Ang II indicate that cardiac function in physiological hypertrophy is either augmented or unaltered whereas it is depressed in pathological hypertrophy.

It is pointed out that the activation of RAS results in the release of two major forms of angiotensin peptides namely Ang II and Ang (1-7) in the circulation [1,2,6,41,42,43]. While both hypertensive and hypertrophic responses of Ang II are elicited by activation of angi-otensin type 1 receptors (AT_1_R), Ang (1-7) has been demonstrated to produce antagonistic effects by activating Mas receptors [6,16,19,41,42,43]. Furthermore, Ang II has been shown to exert antihypertensive and antihypertrophic actions by activating angiotensin type 2 receptors (AT_2_R) [30,44,45,46,47]. Since the activation of both AT_2_R and Mas receptors has been shown to reduce the hypertensive and hypertrophic responses due to AT_1_R activation [18,30,42,43,44,45], the functional significance of AT_2_R and Mas receptors may lie in limiting the development of Ang II-induced cardiac hypertrophy as well as preventing the transition of physiological to pathophysiological cardiac hypertrophy. It is noteworthy that Ang II not only produces hypertrophy of cardiomyocytes but has also been demonstrated to induce marked growth of other types of cells such as fibroblasts in the cardiac extracellular matrix as well as vascular smooth muscles [5,7,9,48,49,50]. Regression of Ang II-induced hypertension, cardiac hypertrophy, and other associated alterations by AT_1_R blocking agents also supports the role of AT_1_R activation in the genesis of Ang II-induced hypertrophic and hypertensive responses [51,52,53,54,55]. This article is intended to provide comprehensive and updated information regarding the functional significance of Ang II-induced adaptive and maladaptive cardiac hypertrophy. In particular, signal transduction mechanisms for the induction of cardiac hypertrophy upon the activation of Ang II receptors will be discussed. In addition, it is planned to describe some therapeutic strategies for the regression of Ang II-induced cardiac hypertrophy.

## 2. Induction of Cardiac Hypertrophy by Angiotensin

Ang II has been documented to induce rapid vasoconstriction, increase cardiac contractility, stimulate myocardial metabolism and produce hypertrophic as well as mitogenic responses [2,5,7,11]. Such effects are also evident upon the activation of both circulating and cardiac RAS and are considered to be elicited by Ang II; however, these actions are independent of each other [1,5,6,8,17,56]. While the activation of RAS for a short period has been shown to produce adaptive cardiac hypertrophy for maintaining cardiovascular function, prolonged activation of RAS or exposure of the heart to Ang II for a prolonged period is known to result in maladaptive cardiac hypertrophy, a well-known risk factor for heart failure [3,4,16,20,26,57]. It should be noted that besides Ang II, different other biologically active peptides such as Ang-(2-8) (Ang III), Ang-(3-8) (Ang IV) and Ang-(1-7) have been shown to exert dramatic effects on the cardiovascular system [9,27]. Furthermore, Ang II-induced actions such as vasoconstriction, cardiac hypertrophy, inflammation, oxidative stress, fibrosis and fluid retention are mediated through Ang II type I receptors (AT_1_R) whereas the effects of Ang-(1-7) are elicited by the activation of Mas receptors [19,25,54,58]. Ang II has also been reported to activate Ang II type II receptors (AT_2_R) and exert actions, which, like the activation of Mas receptors, are antagonists to the effects of AT_1_R activation [2,7,54,58]. It appears that the mechanisms of Ang II for the induction of physiological or pathological cardiac hypertrophy involve the interaction of different types of angiotensin receptors and thus are of complex nature. A schematic representation for the involvement of different types of angiotensin peptides and their receptors is shown in Figure 1. It should be pointed out that under pathological conditions, the activation of circulating RAS by a fall in blood pressure is associated with a release of angiotensinogen from the kidney, which is then converted into Ang I by the action of renin (present in the liver). It has also been demonstrated that angiotensin-converting enzyme (ACE) is involved in the conversion of Ang I to Ang II, Ang III and Ang IV in the lung whereas a homologue of ACE (ACE2) converts Ang I to Ang (1-9) as well as Ang II into Ang (1-7) [7,27]. On the other hand, it has been shown that all these compounds of circulating RAS are present in the local RAS (in various organs), which is activated mainly by an increase in ventricular wall stress [8,9,10]. Although the action of Ang (1-9) is mediated through Ang (1-7), the action of Ang III and Ang IV are considered to be similar to those of Ang II. Nonetheless, the role and significance of Ang (1-9), Ang III and Ang IV are not fully understood at present [27].

Although Ang II has been reported to promote the incorporation ^3^H-leucine and stimulate protein synthesis in cardiomyocytes, fibroblasts and vascular myocytes, the net growth effect is dependent upon the presence of cellular AT_1_R/AT_2_R ratio [47,59,60]. Ang II has been demonstrated to rapidly induce several immediate-early genes such as c-fos, c-jun and c-myc in both myocytes and non-myocytes preparations indicating that the hypertrophic signals by this hormone are similar to those by various growth hormones, which are known to produce physiological hypertrophy [20,35]. On the other hand, upregulation of skeletal alpha-actin, arterial natriuretic factor and other late-gene expression by prolonged exposure of myocytes to Ang II represents markers of pathological hypertrophy and are associated with the fibrosis and inflammation [20,40,48]. It should be mentioned that pathological cardiac hypertrophy by Ang II is also associated with the development of oxidative stress, Ca^2+^-handling defects, apoptosis and autophagy in addition to involving Ca^2+^-calmodulin dependent protein kinases as well as the activation of calcineurin [26,40,48,61]. Furthermore, Ang II-induced hypertrophic and other responses during the occurrence of both physiological and pathological hypertrophy have been reported mainly to be due to the activation of AT_1_R [20,48].

In Ang II induced cardiac hypertrophy, cardiac dysfunction and different signal transduction pathways have been demonstrated to be modified by several pathological factors and conditions. In this regard, it was observed that Ang II did not induce cardiac hypertrophy and produced markedly less degree of apoptosis in transgenic mice with AT_1_R mutation lacking epidermal growth factor receptor transactivation [62]. Ang II also failed to cause hypertension and cardiac hypertrophy in tumor necrosis factor-alpha (TNF-α) knockout mice [28]. Upregulation of M_3_ muscarinic receptors was found to inhibit Ang II-induced cardiac hypertrophy [63]. Both Toll-like receptors (TLR3 and TLR4) were shown to mediate Ang II-induced hypertension and cardiac hypertrophy [3,4]. Some investigators [64] have claimed that the activation of insulin-like growth factor receptors is critical for the induction of hypertension, cardiac hypertrophy and apoptosis by Ang II whereas others have shown Ang II-induced cardiac hypertrophy is attenuated by regulation of autophagy [65]. Both cardiac thyrotropin releasing hormone [21] and increased aldosterone synthase [29] were shown to be required for the development of Ang II-induced cardiac hypertrophy and fibrosis. The involvement of PI3-kinase has been shown in Ang II-induced cardiac hypertrophy due to the formation of oxyradicals, phosphorylation of MAP-kinase and expression of transforming growth factor beta [66,67]. On the other hand, the Wnt/frizzied signaling has been reported to regulate Ang II-induced cardiac hypertrophy upon the activation of glycogen synthase kinase-3 beta [68]. These observations provide evidence that different signal transduction pathways involved in the Ang II-induced of cardiac hypertrophy are affected by several factors and are of complex nature.

## 3. Angiotensin-Induced Signal Transduction Pathways for Hypertrophic Responses

Extensive research work has been carried out over the past 4 decades to understand the hypertrophic effects of Ang II in cardiomyocytes, vascular smooth muscles cells and different types of cells in the extracellular matrix including fibroblasts [5,7,16,18,50,69]. The cellular growth effects of Ang II are associated with various signaling systems including stimulation of phospholipases C and Ca^2+^-mobilization, as well as activation of protein kinase C, MAP kinases, tyrosine kinases and nicotinamide adenine dinucleotide phosphate (NADPH) oxidases (NOX). These Ang II-induced signal transduction alterations are inter-related and associated with increased protein synthesis as well as protein to DNA and RNA to DNA ratios. Furthermore, Ang II actions for cellular growth are mediated by two receptors namely AT_1_R and AT_2_R which are differentially expressed in cardiomyocytes during development. Both AT_1_R and AT_2_R are coupled with G-protein and produce opposing effects [49,70,71]. Although the interaction of AT_1_R and AT_2_R activations are of complex nature, alterations in signal transduction pathways by Ang II are considered physiological under acute conditions for the maintenance of cardiovascular function whereas chronic changes in these mechanisms due to Ang II are associated with pathological situations for the development of cardiac hypertrophy with inflammation, oxidative stress, fibrosis and apoptosis [9,72,73]. The exact reasons for the transition of Ang II-induced physiological to pathological cardiac hypertrophy are not fully understood; however, it appears that the development of a critical level of oxidative stress may be one of the most important pathogenic factors involved in this process.

### 3.1. Ang II-AT_1_R Activated Signaling and Cardiac Hypertrophy

Several studies have revealed that Ang II stimulates phospholipase C (PLC), forming diacylglycerol (DAG) and inositol 1,4,5-triphosphate (IP3) upon the breakdown of phosphatidylinositol 4,5-biphosphate. Subsequently. DAG activates protein kinase C (PKC) and induces hypertrophic effects resulting in cardiac growth [9,25,59,72,74,75,76]. This hypertrophic response to Ang II leads to stimulation of mitogen activated protein kinases (MAPK), including extracellular signal-regulated kinases (ERKs), c-Jun amino-terminal kinases (JNKs), and p38-MAPKs, which potentiates signals transducer and activator of transcription (STAT) pathway as well as other intracellular protein kinases such as non-receptor and receptor tyrosine kinases. It also turns on several downstream signals, such as MAPK/ERK, Ras/Rho, and translocation of MAPK in the nucleus. It is pointed out that different other signaling pathways link the AT1 receptor to Gq-independent phospho-extracellular signal-activated kinase (p-ERK) 1/2 activation by Ang II for cell growth. It is also noteworthy that cardiac hypertrophy has been shown associated with increased concentration of intracellular Ca^2+^ due to the activation of the AT_1_R. In this regard, AT_1_R -Gq/11-phospholipase Cβ (PLCβ) coupling has been reported to produce IP3, which activates IP3 receptor and release Ca^2+^ from the sarcoplasmic reticulum. Activation of sarcolemmal Ca^2+^-channels by AT_1_R has also been shown to increase the concentration of cytosolic Ca^2+^ [12]. However, there seems to be a good correlation between sustained Ca^2+^ release and cell growth indicating that these events may be closely coupled together and in fact, Ca^2+^ has been demonstrated to be required for the development of cardiac hypertrophy by Ang II [6]. A schematic representation of signal transduction pathway involving both protein kinase and Ca^2+^ due to the activation of PLC by AT_1_R is shown in Figure 2.

From the existing information in the literature, it is difficult to sort out the exact signal transduction pathway, which may be responsible for the development of physiological or pathological cardiac hypertrophy by Ang II. However, it appears that the intracellular signaling cascade generated from MAPK constitutes a phosphorylation-based amplification network and results in hypertrophic signals for cardiac adaptive or maladaptive remodeling. Additionally, subfamilies of MAPKs such as p38 kinases, c-JNK, and ERK 1/2, as signaling pathways in cardiac myocytes or extracellular matrix changes have been described to regulate during the progression to the pathological cardiac hypertrophy [77,78,79,80,81,82,83,84]. Furthermore, such changes may be further amplified by the activation of cell membrane CD38, predominant ADP ribosyl (ADPR) cyclase, which is essential for cyclic ADP ribose (cADPR)-mediated intracellular Ca^2+^ mobilization. It may be noted that a marked decrease in Ang-II induced intracellular Ca^2+^ has been reported to occur in CD38 knockdown H9c2 cells; this was associated with a decrease in nuclear factor of activated T cells (NFATc4) translocation and inhibition of ERK/AKT phosphorylation [13]. Likewise, Ca^2+^-dependent signaling proteins such as Ca^2+^/calmodulin-protein kinases and calcineurin are considered to be involved in pathological cardiac hypertrophy because calcineurin dephosphorylates the NFAT transcription factors, promoting nuclear translocation and gene transcription activation. Ca^2+^-calcineurin-NFAT signaling induced hypertrophy is triggered by Ca^2+^mobilizer, cyclic ADP ribose, which is independent of IP3-induced Ca^2+^ release from the sarcoplasmic reticulum by Ang II. Elevated calcineurin activity in human failing ventricular muscle exposed to Ang II has been demonstrated to occur in pathological cardiac hypertrophy [85].

Since Ang II acts as a growth factor to induce cardiac growth, the activation of cardiac proteasome promotes Ang II-induced hypertrophy through the AT_1_R-associated mechanisms. Attenuation of cardiac hypertrophy by proteasome inhibitor (bortezomib) in Ang II infused mice was shown to be due to inhibition of degradation of ATIR-associated proteins and inactivation of AT1R-mediated p38 MAPK and STAT3 signaling pathways; this has been suggested to be beneficial for treating pathological cardiac hypertrophy [86]. In view of the influence of AT_1_R activation on cardiac function by affecting cardiac metabolism, the effects of Ang II on cardiac energy metabolism in experimental models of hypertrophy and diastolic dysfunction have been demonstrated to be associated with marked reduction in cardiac glucose and lactate oxidation without any change in glycolysis or fatty acid β-oxidation in Ang II-treated mice [87]. Since long-term dietary fatty acid intake alters the development of left ventricular hypertrophy, Ang II increased p38 MAPK phosphorylation in rats fed high-fat diet has been specified. Additionally, the increased transcription factor activator protein-1 (AP-1) DNA binding activity in response to Ang II was observed to be higher in rats fed high-oil diet than in those fed standard diet and Ang II downregulated the inducible nitric oxide synthase mRNA levels [88]. Moreover, as a low level of high-density lipoprotein (HDL) is an independent risk factor for pathological cardiac hypertrophy, downregulation of AT_1_R and HDL was shown to ameliorate cardiac hypertrophy via P13K/Akt-dependent mechanism [89]. Ang II-AT_1_R-dependent mechanism has also been reported to induce mechanical stress-triggered pathological cardiac hypertrophy by regulating autophagy [65,89].

The stimulation of AT_1_R induced transactivation of epidermal growth factor receptor (EGFR) has been found to regulate the activation of extracellular signal-activated kinase (ERK) and cardiac hypertrophy in cultured cardiac myocytes [62,79,90]. Additionally, ANG II activated ERK/glycogen-synthase kinase-3 (GSK3), phosphorylated heat shock transcription factor 1 (HSF1), resulting in a protein-coding gene RNF126 (ring finger protein 126) degradation for stabilizing IGF-IIR protein expression and leading to cardiac hypertrophy [91]. Likewise, ANG II activated its downstream kinase JNK, increased IGF-IIR expression through AT_1_R; JNK activation has been shown to degrade sirtuin 1 (SIRT1) via the proteasome and result in heat shock transcription factor 1 acetylation induced IGF-IIR expression for developing cardiac hypertrophy and apoptosis [64]. In another study, the upregulation of the M_3_ muscarinic acetylcholine receptor (M_3_-mAChR) has been indicated during myocardial hypertrophy to relieve the hypertrophic response provoked by Ang II. Furthermore, Ang II-induced M_3_-mAChR overexpression has been demonstrated to attenuate the increased expression of atrial natriuretic peptide and β-myosin heavy chain, and downregulate AT_1_R expression and inhibit the activation of MAPK signaling in the heart [63]. Ang II-induced cardiac hypertrophy in cultured neonatal rat cardiomyocytes was associated with increased visfatin expression mainly through the AT_1_R-JAK/STAT pathway. While an Ang II-induced increase in the expression of visfatin and brain natriuretic peptide was observed in a dose- and time-dependent manner in cardiomyocytes, pre-treatment with AT_1_R antagonist (telmisartan) completely blocked the Ang II-induced visfatin expression increment [92].

Recent studies have explored some novel signaling mechanisms such as a pro-growth factor, Wnt1 inducible signaling pathway protein 1 (WISP1), a target of T-cell factor/lymphoid enhancer factor (TCF/LEF) by which Ang II-AT_1_R promotes cardiac hypertrophy. AT_1_R physical association with NOX2 further enhanced subsequent Ang II stimulation and was associated with increased Akt, p-Akt, p-p38 MAPK, p-ERK1/2, and WISP1 expression [93]. Furthermore, the involvement of small GTP- binding protein Rac has been indicated in Ang-II-induced cardiac hypertrophy [94]. It was demonstrated that the adaptor molecule CIKS is critical in Ang-II-induced cardiomyocyte hypertrophy and is an essential intermediate in Ang-II-induced redox signaling. Ang-II-induced IKK/p65 and JNK/c-Jun phosphorylation, NF-κB, and AP-1 activation have also been reported in cardiac hypertrophy [95]. Thus, it can be appreciated that a wide variety of signal transduction mechanisms are involved in inducing cardiac hypertrophy upon stimulation of AT_1_R by Ang II.

### 3.2. Ang II-AT_I_R/ROS/Redox Signaling and Cardiac Hypertrophy

Hypertrophic stimulus by Ang II can stimulate reactive oxygen species (ROS) formation in cardiomyocytes. Although the formation of Ang II levels at the initial stages for a brief period activates redox-dependent sensitive mechanisms in the heart and contributes to adaptive cardiac hypertrophy, prolonged period of Ang II-induced increased disturbance in the pro-/antioxidants balance due to excess production of ROS in hypertrophied myocardium has shown markedly depressed cardiac function and progression of heart failure [23,61,96,97,98,99,100]. Thus, Ang II-mediated hypertrophic response depends on the increase in low concentration of ROS production which may result in physiological hypertrophy where cardiac function is either unaltered or increased, and this seems to be associated with stimulation of sarcolemmal NOX2. Since the expression of AT_1_R is redox dependent, the overproduction of ROS results in the overstimulation of AT_1_R-mediated pathways for a prolonged period leading to oxidative stress. These effects of chronic Ang II exposure result in mitogenic, proinflammatory, and profibrotic actions causing hypertrophic cell growth, cardiac remodeling, and pathological cardiac hypertrophy [100,101,102,103]. Excessive ROS production via different types of NOX disrupts redox signaling within the cells and is considered to induce pathological growth of cardiac myocytes [97,98,100,104,105].

NOX activation during early response of endothelial cells to Ang II by binding to the AT_1_R, is essential for the formation of ROS in various cardiovascular cell types. Augmented NOX activity is a source of induction of ROS that has been implicated in the development of pathological hypertrophy. Ang II-induced hypertrophic effects contribute to ERK1/2, Akt, and NF-kB signaling via NOX-dependent ROS formation, and subsequent activation of p38 MAPK, c-JNK, and nuclear factor-κB (NF-κB) as an essential mechanism which induces cardiomyocyte hypertrophy [100,101,106,107,108]. Activation of the PKC-ERK-NF-κB signaling pathway and increased intracellular ROS induced cardiomyocyte hypertrophy by regulating expression levels of NOX2 and NOX4 has been demonstrated. As NF-κB is an oxidative sensitive transcriptional factor, Ang II-AT_1_R activation of NOX2 has been reported to increase ROS contribution in inducing hypertrophic effects and involvement of ERK1/2, Akt, and NF-κB signaling. A schematic representation of NOX2 and NOX4 in the development of physiological cardiac hypertrophy and pathological cardiac hypertrophy is shown in Figure 3.

It should be mentioned that NOX2 is the predominant protein identified in cardiac sarcolemma and transduces downstream signaling events for seven NOX known isoforms. Low levels of ROS are produced by NOX2 for physiological processes such as cell proliferation, migration, differentiation, and cytoskeletal organization, whereas excessive production of ROS from the activated NOX4, which is mainly localized in mitochondria, contributes to pathological cardiac hypertrophy [109,110,111]. Ang II-stimulated ROS generation via NOX in cardiomyocytes is supported by the blockade of gp91phox-NOX2, which attenuated Ang II-induced cardiac hypertrophy [94]. Additionally, cardiac-specific overexpression of NOX4 in mice potentiated Ang II-induced cardiac hypertrophy, which is inhibited by GKT137831 administration. The mechanisms involved include upregulation of NOX4 levels, NOX4-dependent ROS production, and increased phosphorylation of RACα serine/threonine-protein kinase (Akt). Phosphorylation of the two downstream effectors of Akt, mechanistic target of rapamycin (mTOR) and NF-κB, specifically, the p65 subunit were found to be upregulated in the hearts of Ang II-infused mice. In this model of transient overexpression of NOX4 in the heart, NOX4-induced exacerbated Ang II-cardiac hypertrophy via increased ROS production has been reported [108]. Moreover, since cardiomyocyte enlargement is the most defining characteristics of cardiac hypertrophy, Ca^2+^-dependent NOX5 was observed to exaggerate cardiac hypertrophy through ROS production. Augmented Ang II-induced cardiomyocyte enlargement accompanied by significant increases in the fetal genes ANP and β-MHC have also been demonstrated [112,113,114].

### 3.3. Ang II-AT_1_R Induced ROS—Mitochondrial Dysfunction

It has been indicated that mitochondrial dysfunction is a significant source of ROS, and elevated mitochondrial ROS formation is involved in Ang II-induced pathological cardiac hypertrophy. Indeed, several studies have shown that Ang II enters mitochondria and stimulates NOX4, promotes electron leak and mitochondrial ROS production; ROS produced by NOX4 also causes mitochondrial DNA damage, oxidation of components of the membrane permeability transition pore, and opening of the mitochondrial ATP-sensitive K^+^ channels [115,116,117,118]. Inhibition of mitochondrial ROS production by SS-31 or genetic transfer of catalase targeted to mitochondria was found to prevent Ang II-induced cardiac hypertrophy, and diastolic dysfunction in mice [119]. In Ang II-infused animals, ROS scavenging with N-acetylcysteine was less effective than mitochondria-targeted scavenging with peptide SS-31 in preventing cardiac hypertrophy, suggesting that mitochondrial ROS has an essential role in modulating cardiac remodeling in Ang II-infused animals [120]. Furthermore, mitochondrial cyclophilin D, which acts as a Ca2+ sensitizer for mitochondrial permeability transition pore opening, mediates Ang II-induced mitochondrial superoxide production [121,122]. Likewise, this agent altered mitochondrial function in vivo in Ang II-infused mice and this supports the view that cardiac hypertrophy is associated with reductions in cardiac glucose oxidation and ATP production. There also occurs an upregulation of pyruvate dehydrogenase kinase 4 via activation of the cyclin/cyclin-dependent kinase-retinoblastoma protein-E2F pathway in response to Ang II [87]. Since Ang II-induced mitochondrial metabolic shift is considered a cause of cardiac hypertrophy, Ang II infusion has been shown to reduce cardiac fatty acid oxidation associated with enhanced glycolysis. These effects were reversed by inducible cardiac-specific deletion of acetyl CoA carboxylase, and the associated cardiac hypertrophy was improved [123]. In fact, NOX-dependent uncoupling of eNOS and consequent mitochondrial dysfunction resulting in sustained oxidative stress is an effective mechanism for developing pathological cardiac hypotrophy. It is also increasingly evident that increased ROS and oxidative stress also result from the activities of endogenous antioxidants such as superoxide dismutase, glutathione peroxidase, and catalase in Ang II-induced cardiac hypertrophy [124,125,126].

### 3.4. Ang II-AT_1_R Induced ROS and Nuclear Factor Erythroid-2 Elated Factor 2 (Nrf2)

Nrf2 is known as an essential regulator of ROS formation in cardiomyocytes, and it has been indicated that increased ROS generation and PI3K-Akt signaling activate the receptor Nrf2. Since Nrf2 has a critical role in antioxidant defenses, Nrf2 knockout has been shown to enhance Ang II-induced cardiac hypertrophy by further increasing oxidative stress in the heart. It has also been demonstrated that Nrf2 is a novel negative regulator of Ang II-mediated cardiomyocyte hypertrophy and maladaptive cardiac hypertrophic partly via the suppression of oxidative stress, independent of changes in blood pressure [83,127,128,129,130]. It has recently been documented that Nrf2 deficiency exacerbates Ang II-induced cardiac hypertrophy via oxidative stress-dependent down-regulation of p27kip1 [128]. In contrast, activation of Nrf2 was shown to suppress the axis of Ang II-oxidative stress in cardiomyocyte hypertrophy; exacerbated cardiomyocyte hypertrophy induced by Ang II due to Nrf2 deficiency has also been shown in Nrf2 KO mice [83]. As a master transcription factor expressed in most tissues, Nrf2 exhibits a significant role in amplifying the antioxidant pathways associated with the enzymes present in the myocardium as is significantly engaged in regulating the gene expression of oxidants and antioxidants by binding with antioxidant response elements [131]. It has also been reported that astragaloside IV improved cardiac hypertrophy and LV function and structure as well as increased expression of Nrf2 and heme oxygenase-1 has been shown [132]. These observations are consistent with the view that depression in Nrf2 activity may reduce the antioxidant reserve in the myocardium and such a change may be responsible for the progression of pathological cardiac hypertrophy.

### 3.5. Ang II-AT_2_R and Ang (1-7)-Mas Receptor Activated Signaling Mechanisms in Cardiac Hypertrophy

The cardioprotective effects of AT_2_R activation by counteracting the effects of AT_1_R in Ang II-induced cardiac hypertrophy were evident as the blockade of AT_2_R stimulation was demonstrated to augment the early signals of AT_1_R-mediated cardiac growth responses [133,134,135,136,137,138]. It is pointed out that AT_2_R belongs to the family of GPCRs with various downstream signaling mechanisms depending on the cell type. AT_2_R associated signaling mechanisms involved in the inactivation of AT_1_R–activated MAPK protein tyrosine phosphatase stimulation, prevention of thyroid hormone-induced cardiac mass gain, and activation of Akt have been revealed [139]. Ang II- AT_1_R activation elevated Ca^2+^ levels and PKC activation have also been indicated upon downregulating the AT_2_R expression in cardiac myocytes [30]. Furthermore, the activation of AT_2_R has been shown to promote vasorelaxation through PKA-dependent eNOS activation and paracrine signaling through bradykinin/cGMP/NO production [137]. AT_2_R has also been reported to activate the kinin/NO/cGMP system and protein tyrosine phosphatase as well as serine/threonine phosphatase stimulation [140]. By binding to AT_2_R, Ang II antagonize the effect of AT_1_R by promoting vasodilation through NO and cGMP stimulation, anti-proliferation, natriuresis, antiangiogenesis, antifibrosis, and anti-inflammation in various tissues, including endothelium, vascular smooth muscle, heart, brain, and kidney [69,73,138]. Since infusion of Ang-II in mice lacking the AT_2_R gene did not show any development of cardiac hypertrophy, it was suggested that AT_2_R signaling pathway may participate in the development of Ang-II-induced cardiac hypertrophy [141]. Primarily dependent on the AT_2_R, Ang II was found to upregulate expression and secretion of a potential myocardial hypertrophy factor cyclophilin A through ROS production in rat cardiomyocytes [142]. Significant increase in the level of AT_2_R expression and contribution of AT_2_R in the activation of Akt have also been observed in the development of the thyroid hormone-induced cardiac hypertrophy [143]. Although these observations suggest a dual role of Ang II-AT_2_R activation in the hypertrophic process depending upon the type and stage of cardiac hypertrophy, most of the studies favor its antihypertrophic effect in regulating cardiac hypertrophy due to AT_1_R activation.

Ang (1–7), one of the significant enzymatic products of ACE2, has been shown to attenuate Ang II-induced pathological cardiac hypertrophy by its cardioprotective effects mediated by Mas receptors through different signaling pathways. However, there is evidence to suggest that Ang (1–7) may promote signaling via Mas receptors in a G protein-independent manner in spite of the fact that Mas receptors have solid constitutive activities with Gq and G12. Most studies have shown that A (1–7) or other Mas agonists in the heart induce antihypertrophic and cardioprotective effects [143,144,145,146,147]. Treatments of cardiomyocytes with Ang(1–7) have been shown to attenuate Ang II-induced cardiac hypertrophy [143,147,148]. Furthermore, acute exposure to Ang(1–7) in cardiomyocytes did not show any noticeable effect on Ca^2+^ transients but promoted NO release by activating endothelial NO synthase (eNOS) and nNOS. Alternatively, significant effects on Ca^2+^-handling proteins upon chronic exposure to Ang(1–7) or genetic deletion of Mas receptors have been reported. Additionally, Ang-(1–7)-producing fusion protein in the heart showed an increased Ca^2+^ transient amplitude, faster Ca^2+^ uptake, and increased expression of SERCA2 [143,149]. Recent work also points to protective functions of Ang (1–7)/AT_2_R signaling as Ang (1–7) was shown to mediate vasodilation via AT_2_R in the presence of an AT_1_R blocker [150]. However, it needs to be pointed out that the current knowledge of Ang (1–7)/Mas receptors signal transduction in cardiac hypertrophic processes is limited, and more experimental and clinical research for the understanding of its mechanisms is required [151,152].

## 4. Therapeutic Strategies for Preventing or Regression of Ang II-Induced Cardiac Hypertrophy

Since Ang II stimulates cardiovascular growth and remodeling by binding to AT_1_R, many AT_1_R blockers such as losartan, valsartan, telmisartan and candesartan have shown to attenuate cardiac hypertrophy [153,154,155,156]. It is pointed out that AT_1_R blockade has not only been shown to prevent the development of Ang II-induced cardiac hypertrophy but also known to promote its regression. Several studies have reported the antihypertrophic effect of different synthetic and natural compounds such as curcumin and resveratrol by inhibiting some target sites in various signal transduction pathways in Ang II-induced cardiac hypertrophy [60,61,66,67,157,158]. Furthermore, curcumin, losartan, and anti-LOX-1 antibodies were found to attenuate Ang II-mediated oxidative stress, the expression of NOX and NF-κB as well as cardiac hypertrophy [157]. Attenuated activation and expression of AT_1_R upon inhibiting the phosphorylation of PKC-ERK-NF-κB pathway by Pterosin B have been shown to exert beneficial effects [60]. Reduction in excessive intracellular ROS by Pterosin B for regulating the expression levels of NOX2 and NOX4 has also been demonstrated to attenuate Ang II-induced cardiomyocyte hypertrophy [159]. Liraglutide was also shown to ameliorate cardiac hypertrophy potentially by suppressing the AT_1_R-mediated events and preventing the progression of cardiac hypertrophy to heart failure [160].

Pre-treatment of neonatal cardiomyocytes by an AT_2_R blocker PD123319, was demonstrated to increase the hypertrophic effects of AT_1_R activation by Ang II whereas the antigrowth effects of AT_2_R activation by Ang II became more evident upon treatment with an AT_1_R blocker, losartan. Accordingly, it was suggested net growth effect of Ang II depends on the cellular AT_1_/AT_2_ receptor ratio [60]. Additionally, AT_2_R blockade was shown to prevent thyroid hormone-induced cardiac mass gain and Akt activation, indicating the role of AT_2_R in developing future therapeutic strategies for the treatment of pathological cardiac hypertrophy [139]. Apart from AT_1_R blockers, ACE inhibitors such as enalapril, ramipril, benazepril, zofenopril, lisinopril, fosinopril, perindopril, and imidapril which reduce the formation of Ang II, have been evidenced for their beneficial effects in attenuating pathological cardiac hypertrophy [27,161,162,163,164,165,166,167]. In fact, the combination of ACE inhibitors with AT_1_R blockers, eprosartan has been reported to improve cardiac output in patients with severe heart failure [167]. Although several ACE inhibitors and AT_1_R antagonists are used clinically for the prevention or reversal of pathological cardiac hypertrophy and subsequent heart failure, it remains to be investigated whether these beneficial effects are associated with elevations in the level of Ang (1-7) or activities of Mas receptors and AT_2_R. Because Ang II is known to produce oxidative stress, it has been suggested that the antihypertrophic effects of ACE inhibitors and AT_1_R blockers may be due to the antioxidant activities. The use of different antioxidants has shown to reduce pathological cardiac hypertrophy as well as vascular remodeling [168,169]. In this regard, scoparone was reported to inhibit Ang II-induced cardiac hypertrophy in vitro via the elimination of overexpression of RAC1 and by inhibiting RAC1-mediated oxidative stress [170,171].

Since the vital role of endogenous antioxidant defenses in the control of Ang II-mediated redox signaling in the heart, up-regulation of antioxidant enzymes (such as haeme-oxygenase-1 and thioredoxin 2), have been demonstrated to inhibit Ang II-induced oxidative stress and cardiac hypertrophy [172,173,174]. The activators of NF-κB such as astragaloside can be seen to exert beneficial effects for preventing the Ang II- induced pathological hypertrophy by elevating the level of antioxidant reserve in cardiomyocytes [132]. Additionally, activation Rac1-a significant regulator of NOX activity in adult hearts, is required for Ang II-induced cardiac hypertrophy [175,176]. In cardiomyocytes and cardiac fibroblasts, Ang II activated Rac1 (increasing expression of RAC1-GTP), as well as NOX2 and NOX4 involvement in cardiac hypertrophy and fibrosis has been indicated [174,177]. The antioxidants therapies are considered to have an advantage for the treatment of cardiac hypertrophy over ACE inhibitors or AT_1_R blockers [17,27,57,61,174,178]. Furthermore, it is suggested that extensive effort should be made to develop appropriate activators of AT_2_R and Mas receptors for preventing or reversing the Ang II- induced pathological cardiac hypertrophy [133,134,143,149].

## 5. Concluding Remarks

From the foregoing discussion, it is evident that induction of cardiac hypertrophy by elevated levels of Ang II upon the activation of RAS is considered to maintain cardiovascular function. Although Ang II is also known to produce hemodynamic overload due to vascular vasoconstriction, cardiac hypertrophy induced by this hormone has been shown to be independent of pressure overload. Nonetheless, acute exposure of the heart to Ang II has been reported to produce physiological cardiac hypertrophy with augmented or unaltered cardiac function whereas chronic exposure results in pathological cardiac hypertrophy with depressed cardiac function. Extensive research work has revealed that the hypertrophic response of cardiomyocytes, vascular myocytes and the other cell types in extracellular matrix including fibroblasts to Ang II is primarily elicited by the activation of AT_1_R whereas the activation of AT_2_R results in antihypertrophic effects. Both AT_1_R and AT_2_R are coupled with different signal transduction molecules through Gq-proteins. The activation of Mas receptors by another angiotensin peptide, Ang-(1-7), has also been shown to exert antihypertrophic response of the myocardium. It is becoming apparent that the net growth of myocardium due to Ang II and subsequent stimulation of signal transduction pathways is a consequence of the activation of AT_1_R and AT_2_R or Mas receptors. It is also clear that an increase in the concentration of intracellular Ca^2+^ is absolutely essential for the activation of AT_1_R-linked signal transduction mechanisms for the induction of cardiac hypertrophy by Ang II.

Although a wide variety of signal transduction pathways are involved during the development of cardiac hypertrophy by Ang II, it is difficult to clearly identify which one is associated with physiological or pathological hypertrophy. Such a complexity in identifying the role of any one pathway appears to be related the fact that all these signal transduction mechanisms are closely inter-related to each other for the occurrence of oxidative stress, inflammation and intracellular Ca^2+^-overload. There is ample evidence to suggest that the hypertrophic response of cardiomyocytes to Ang II involves the AT_1_R-induced activation of PLC for the formation of DAG and IP3. Furthermore, there occurs the activation of PKC by DAG, which then results in the development of cardiac hypertrophy as a consequence of the activation of ERK1/2. On the other hand, IP3 formed due to the activation of PLC results in the activation of Ca^2+^-calmodulin kinase by binding with IP3 receptors in the sarcoplasmic reticulum and release Ca^2+^ in the cytoplasm. Since the activation of AT_1_R has also been demonstrated to increase Ca^2+^-entry through Ca^2+^-channels in the sarcolemmal membrane, this mechanism can also be seen to participate in raising the intracellular Ca^2+^ and subsequent cardiac hypertrophy.

Another signal transduction mechanism for the induction of cardiac hypertrophy by Ang II involves the production of ROS. It appears that the stimulation of AT_1_R upon acute exposure of myocardium to Ang II activates NOX2 and produces low concentrations of ROS for altering the redox status of myocardium. This change has been shown to be associated with the activation of Akt, increased activity of NF-kB and activation of ERK1/2 for the occurrence of physiological cardiac hypertrophy. On the other hand, chronic exposure of myocardium to Ang II stimulates NOX4 upon the activation of AT_1_R and results in the development of oxidative stress due to excessive production ROS. Such a change has been shown to produce intracellular Ca^2+^-overload and activate calcineurin for inducing pathological cardiac hypertrophy as a consequence of the occurrence of inflammation, apoptosis and fibrosis. It is suggested that both these signal transduction mechanisms involving oxyradical formation and PLC activation may participate in the genesis of Ang II-induced cardiac hypertrophy. It is pointed out that it is not our intention to rule out the role of several other pathways and growth factors in the hypertrophic process due to Ang II. Thus, various interventions inhibiting AT_1_R or activating AT_2_R and Mas receptors as well as affecting different signal transduction pathways can be seen to produce beneficial effects in reducing the oxidative stress, inflammation, and intracellular Ca^2+^- overload for preventing the Ang II-induced cardiac hypertrophy.

## Figures and Tables

**Figure 1 cells-11-03336-f001:**
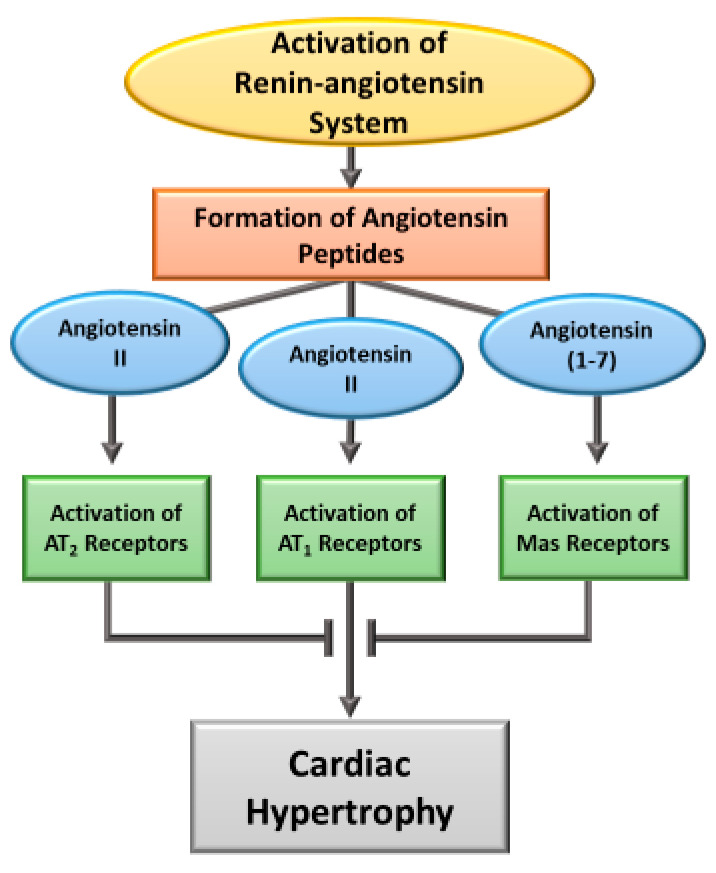
A schematic representation for the involvement of different angiotensin peptides such as Ang II and Ang (1-7) as well as their receptors for the development of cardiac hypertrophy upon the activation of renin-angiotensin system. It is pointed out that although other angiotensin peptides such as Ang III, Ang IV and Ang (1-9) are also formed during the activation of renin-angiotensin system, their role and receptor mechanisms are not fully understood.

**Figure 2 cells-11-03336-f002:**
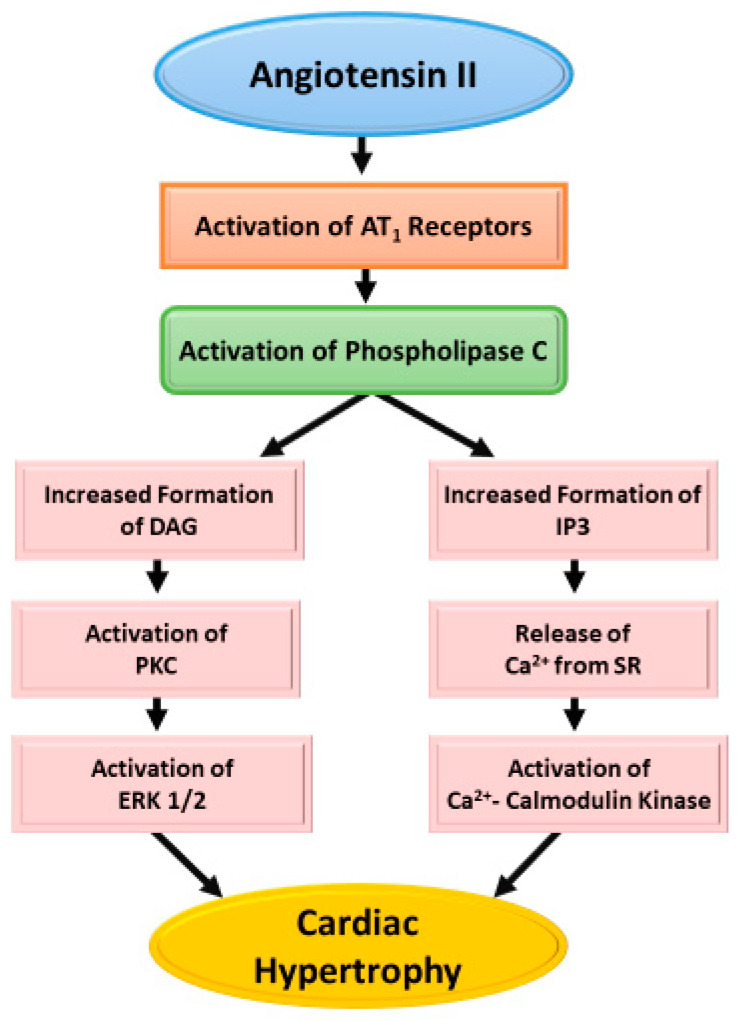
A schematic representation for the AT_1_ receptors and phospholipase C mediated signal transduction pathways for the development of Angiotensin II- induced cardiac hypertrophy. The activation of AT_1_R has been shown to promote Ca^2+^—entry through sarcolemmal Ca^2+^ channels and increase the intercellular concentration Ca^2+^, which may also contribute to activating Ca^2+^ calmodulin kinase for the occurrence of cardiac hypertrophy.

**Figure 3 cells-11-03336-f003:**
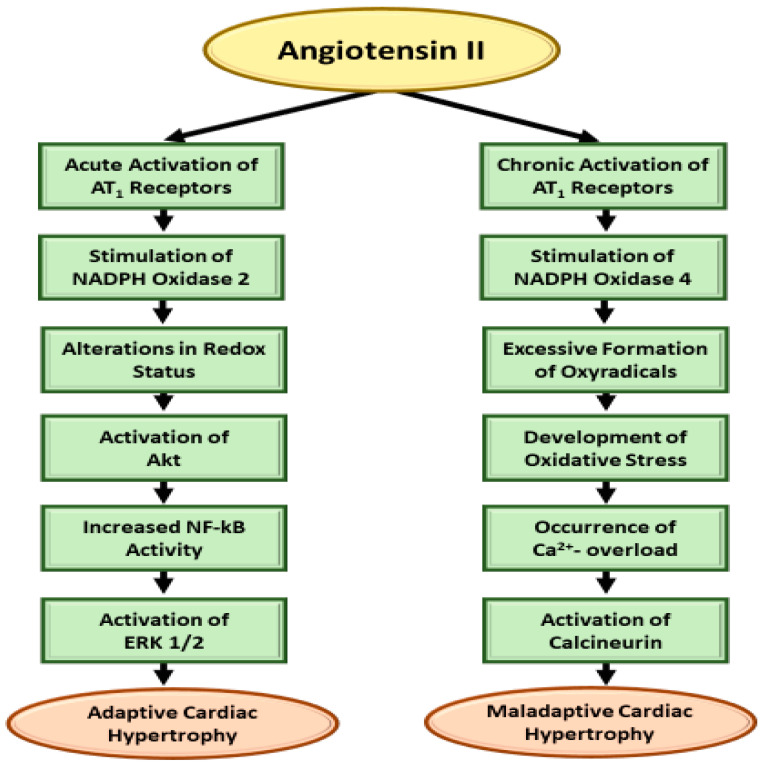
A schematic representation for the AT_1_ receptors and redox status mediated signal transduction pathway as well as AT_1_ receptor and oxidative stress-calcineurin pathway for the development of acute and chronic cardiac hypertrophy, respectively. Although different other signal transduction pathways such as ROS, PKC, ERK1/2, Akt, NF-κB and NOX, p38 MAPK, c-JNK, NF-κB have been identified to explain Ang II—induced pathological (maladaptive) cardiac hypertrophy, their involvement including that of Ca^2+^, calmodulin in the development of physiological (adaptive) cardiac hypertrophy is poorly understood.

## Data Availability

Not applicable.

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
