# Peer review of "Angiotensin II-Induced Signal Transduction Mechanisms for Cardiac Hypertrophy"

_cells, 2022, doi:10.3390/cells11213336_

Round 1

Reviewer 1 Report

This review article by Bhullar and Dhalla provides a detailed summary of angiotensin II (AngII) signaling and AngII-mediated regulation of cardiac hypertrophy. The authors nicely describe established signaling paradigms associated with AngII induction of growth responses within cardiomyocytes and provide an overview of the cellular and organ level impact on physiological and pathological heart growth. The paper is well written, broad enough to encompass a heavily researched field, yet provides enough explanation to ensure clear communication of the most pertinent findings and mechanisms. I only have a few suggestions that may further strengthen the review.

1.     It may be helpful to the reader if the authors briefly discuss how AngII, Ang1-7, and other potential products are generated by RAS during physiological and pathological conditions.

2.     Perhaps Figure 2 could be more inclusive of signaling pathways elicited by AngII binding and activation of AT1R. several additional signaling pathways are described in the review, e.g. NOX enzymes, calcineurin, RAC1, etc., and could be visualized in a broader schema here.

3.     The flow of the review could be enhanced by moving the AngII-AT2R and Ang1-7-Mas sections after subsection 3.5. This would ensure all AngII-AT1R signaling is uninterrupted.

4.     In the Therapeutic Strategies, it might be helpful to point out which interventions are currently used clinically.

Author Response

Reviewer #1: 

  1. We have made a computer check of the revised manuscript and have corrected spelling mistakes.
  2. We have now included a few statements to indicate the formation of different forms of Ang products generated by RAS in Section 2 of the revised manuscript.
  3. We have now added a statement in the legend for Figure 2 to emphasis a separate route of Ca2+-entry due to the activation of AT1
  4. We have now moved the subsection of 3.2 to 3.5 to ensure that all Ang II-AT1R signaling is uninterrupted. This has resulted in the change of references # (96-115) to # (133-152) as well as reference # (116-153) to # (96-132). These reference # have been corrected in the subsection 3.5 as well as Reference Section.
  5. Interventions being used both experimentally and clinically are now clearly indicated in the Section #4.

Reviewer 2 Report

This review manuscript provides a concise and comprehensive overview of the role angiotensin II signaling in the development of cardiac hypertrophy, with an emphasis on the signaling mechanisms of Ang II-induced physiological v.s. pathological cardiac hypertrophy via AT1R and AT2R, respectively. A large number of signaling molecules and pathways have been discussed that interact with and participate in Ang II signal transduction, with variable pathophysiological consequences. Strategies to prevent and reverse Ang II-induced cardiac hypertrophy have also been discussed. Overall, the manuscript is well written.

It would be desirable to make the figures more comprehensive, interactive and informative.

Author Response

We have depicted some of events and signal transduction pathways for Ang II-induced physiological and pathological cardiac hypertrophy in the Figures 1 to 3. Since the signal transduction pathway for Ang II-induced two forms of cardiac hypertrophy are not only complex in nature but are also inter-related, we do not wish to make these novel and original Figures too confusing. Accordingly, we have added appropriate statements regarding some additional pathways in the legends to all these three Figures to make these readily more comprehensive, interactive and informative. I hope this is acceptable.